# Effect of the Cannabinoid Agonist WIN 55,212-2 on Neuropathic and Visceral Pain Induced by a Non-Diarrheagenic Dose of the Antitumoral Drug 5-Fluorouracil in the Rat

**DOI:** 10.3390/ijms241914430

**Published:** 2023-09-22

**Authors:** Gema Vera, Laura López-Gómez, Rocío Girón, María Isabel Martín-Fontelles, Kulmira Nurgali, Raquel Abalo, José Antonio Uranga

**Affiliations:** 1Department of Basic Health Sciences, Faculty of Health Sciences, University Rey Juan Carlos (URJC), 28922 Alcorcón, Spain; gema.vera@urjc.es (G.V.); laura.lopez.gomez@urjc.es (L.L.-G.); rocio.giron@urjc.es (R.G.); isabel.martin@urjc.es (M.I.M.-F.); jose.uranga@urjc.es (J.A.U.); 2High Performance Research Group in Physiopathology and Pharmacology of the Digestive System (NeuGut-URJC), University Rey Juan Carlos (URJC), 28922 Alcorcón, Spain; 3Associated I+D+i Unit to the Institute of Medicinal Chemistry (IQM), Scientific Research Superior Council (CSIC), 28006 Madrid, Spain; 4High-Performance Research Group in Experimental Pharmacology (PHARMAKOM-URJC), University Rey Juan Carlos (URJC), 28922 Alcorcón, Spain; 5Institute for Health and Sport, College of Health and Biomedicine, Victoria University, Melbourne, VIC 3011, Australia; kulmira.nurgali@vu.edu.au; 6Department of Medicine Western Health, University of Melbourne, Melbourne, VIC 3010, Australia; 7Regenerative Medicine and Stem Cells Program, Australian Institute of Musculoskeletal Science (AIMSS), Melbourne, VIC 3021, Australia; 8Working Group of Basic Sciences on Pain and Analgesia of the Spanish Pain Society, 28046 Madrid, Spain; 9Working Group of Cannabinoids of the Spanish Pain Society, 28046 Madrid, Spain

**Keywords:** cancer chemotherapy, cannabinoid, chemotherapy-induced adverse effects, gastrointestinal motor function, peripheral neuropathy, visceral pain

## Abstract

5-fluorouracil (5-FU) is an antineoplastic drug used to treat colorectal cancer, but it causes, among other adverse effects, diarrhea and mucositis, as well as enteric neuropathy, as shown in experimental animals. It might also cause neuropathic pain and alterations in visceral sensitivity, but this has not been studied in either patients or experimental animals. Cannabinoids have antimotility and analgesic effects and may alleviate 5-FU-induced adverse effects. Our aim was to evaluate the effects of the cannabinoid agonist WIN 55,212-2 on neuropathic and visceral pain induced by a non-diarrheagenic dose of 5-FU. Male Wistar rats received a dose of 5-FU (150 mg/kg, ip) and gastrointestinal motility, colonic sensitivity, gut wall structure and tactile sensitivity were evaluated. WIN 55,212-2 (WIN) was administered to evaluate its effect on somatic (50–100 µg ipl; 1 mg/kg, ip) and visceral (1 mg/kg, ip) sensitivity. The cannabinoid tetrad was used to assess the central effects of WIN (1 mg/kg, ip). 5-FU decreased food intake and body weight gain, produced mucositis and thermal hyperalgesia, but these effects were reduced afterwards, and were not accompanied by diarrhea. Tactile mechanical allodynia was also evident and persisted for 15 days. Interestingly, it was alleviated by WIN. 5-FU tended to increase colonic sensitivity whereas WIN reduced the abdominal contractions induced by increasing intracolonic pressure in both control and 5-FU-treated animals. Importantly, the alleviating effects of WIN against those induced by 5-FU were not accompanied by any effect in the cannabinoid tetrad. The activation of the peripheral cannabinoid system may be useful to alleviate neuropathic and visceral pain associated with antitumoral treatment.

## 1. Introduction

5-Fluorouracil (5-FU) is a pyrimidine analogue that exerts a non-competitive inhibitory effect of thymidylate synthase, an enzyme necessary for DNA replication [1]. 5-FU has been used to treat numerous tumors, including colon, breast, and head and neck carcinomas [2,3]. Unfortunately, 5-FU induces adverse side effects such as diarrhea, mucositis, neutropenia, and vomiting [4,5], which can remain up to 10 years post-treatment [6,7].

5-FU-induced gastrointestinal dysfunction is associated with inflammation, epithelial destruction, and intestinal ulceration causing intestinal mucositis [8,9]. In humans, mucositis can be followed by clinical manifestations such as diarrhea and abdominal pain [10,11]. In animal models, even after inflammation resolution, 5-FU-induced gastrointestinal dysmotility can persist [12] and is associated with the development of an enteric neuropathy [13]. 

Other side effects such as neurotoxicity affecting the peripheral nervous system, manifested as somatic neuropathic pain, are less clearly associated with the use of 5-FU. The development of peripheral neuropathy has been more often related with other antitumoral drugs like vincristine, cisplatin or oxaliplatin [14,15,16]. Thus far, only a limited number of studies have demonstrated that 5-FU induces neuropathic pain, and in all cases, it was administered in combination with other chemotherapy agents (platinum derivatives, taxanes or even radiotherapy) [17,18]. To our knowledge, animal studies have only demonstrated enteric [13], but not somatic peripheral, neuropathy.

Visceral pain is difficult to localize, irradiates to superficial structures, and often occurs with nausea, vomiting, and other manifestations [19,20]. Even though these effects are often found in patients treated with chemotherapy, visceral pain associated with antitumoral treatment has received scarce attention, especially in experimental models [21,22]. Visceral pain might be related with other side effects demonstrated to occur with chemotherapy like mucositis [23,24] and enteric neuropathy [25,26]. To our knowledge, thus far, no work has studied the effect of 5-FU on visceral sensitivity.

Despite the general concerns regarding their central effects, cannabinoids are widely used to prevent nausea, vomiting, and pain, and to increase appetite in patients with cancer treated with chemotherapeutic agents [27,28,29]. 

In a previous work performed in rats, the non-selective cannabinoid agonist WIN 55,212-2 (WIN) at a non-psychotropic dose partly attenuated 5-FU-induced diarrhea, but did not counteract the associated mucositis [30]. Interestingly, this and other cannabinoids may improve peripheral tactile neuropathy induced by different antitumoral drugs in rodents, including cisplatin, paclitaxel, vincristine and oxaliplatin [31,32,33,34,35,36,37]. However, their effects on the alterations of somatic and visceral sensitivity that 5-FU may cause have not yet been studied. 

Therefore, the aims of this work were:To characterize in the rat the effects of a relatively low dose of 5-FU on body weight, food and water intakes, gastrointestinal motor function, tactile mechanical and thermal sensitivity, colonic mechanical sensitivity, and the general structure of the gut (ileum, colon) wall;To evaluate the effects of WIN on the somatic and visceral nociceptive thresholds altered in the rat by 5-FU. In addition, the possible central effects of WIN, which could be seen as a drawback of cannabinoid treatment, were analyzed.

## 2. Results

### 2.1. General Health Parameters

The effects of 5-FU at 150 mg/kg on general health parameters can be seen in Figure 1. The antitumoral drug reduced body weight acutely (days 3–4 after treatment administration; *p* < 0.01) and this decrease was maintained at least during the first 10 days after treatment. However, at the end of the study (15 days after 5-FU administration), body weight reached similar values to those of control animals (Figure 1A). 

Also, the antitumoral drug reduced food intake acutely on the first day after treatment (*p* < 0.05), but thereafter (days 5–15), saline and 5-FU-treated animals ingested the same amount of food (Figure 1B). 

Finally, water intake was not significantly modified during 5-FU treatment (Figure 1C).

### 2.2. Gastrointestinal Motor Activity

Gastrointestinal motility was evaluated using radiographic methods 4 and 14 days after 5-FU (or saline) administration. As shown in Figure 2, 4 days after drug administration, the gastrointestinal motor patterns were similar in control and 5-FU-treated animals (Figure 2A–E). Moreover, 14 days after 5-FU administration, no relevant modifications were found in the motility patterns of the different gastrointestinal regions between control and 5-FU-treated rats either (*p* > 0.05).

Interestingly, 5-FU at this dose did not cause observable diarrhea; feces collected from the cages at different time points showed normal appearance, and no evident sign of diarrhea on the perianal region of the rats was found.

### 2.3. Gut Wall Histology

As seen in Figure 3, 5-FU treatment caused histological damage to the overall structure of the ileum and colon 4 days after drug administration, although the difference with saline-treated animals only reached statistical significance in the small intestine. However, 15 days after drug administration, the values were normalized in both organs and the preparations looked similar to the control animals treated with saline.

Ileal damage on day 4 specifically manifested in the presence of enlarged lymph vessels (Figure 3B, insert; Figure 4B, arrows) and in a significant decrease in the height of intestinal villi (Figure 4A–C). Furthermore, there was an increase in the immune infiltration of the mucosa, the number of chromogranin-immunoreactive enteroendocrine cells in the epithelium (Figure 4D–F), the number of neurons per myenteric ganglion (Figure 4G–I), and the thickness of the external muscular layers (Figure 4J–L). Of these differences, only that affecting the myenteric neurons reached statistical significance compared with control rats, and only that affecting enteroendocrine cells was not clearly corrected with time.

### 2.4. Development of Peripheral Neuropathy and Effect of WIN

In control animals, the threshold for mechanical sensitivity in the von Frey test was 24.53 ± 1.19 g, and this value did not significantly change at 15 days, whereas 5-FU administration induced peripheral neuropathy manifested as a significant reduction in the threshold for mechanical sensitivity. This effect could be appreciated 3 days after 5-FU injection, and it remained for at least 15 days after administration (Figure 5A). The latency of paw withdrawal in the plantar test was also reduced 3 days after 5-FU administration, but contrary to mechanical allodynia, this sign of thermal hyperalgesia was not present any more on day 15 of the experiment (Figure 5B).

The effect of WIN on somatic sensitivity was tested after its local and systemic administration.

As the effect of 5-FU on thermal sensitivity did not remain for the whole experiment, the effect of the locally administered cannabinoid is only shown for the von Frey hair test. For this, WIN was administered at two different doses, 50 and 100 µg. At the lowest dose used, WIN was not able to reverse the mechanical allodynia induced by 5-FU administration in the rats (Figure 5C). However, 100 µg was able to alleviate the mechanical allodynia induced by 5-FU, at least partially (Figure 5D).

Finally, in 5-FU treated rats, the acute intraperitoneal administration of WIN at 1 mg/kg reversed the effect of the drug on the threshold for mechanical sensitivity, almost reaching the typical values obtained for control rats (Figure 5E).

### 2.5. Cannabinoid Tetrad

In order to detect if systemically administered WIN was able to cause psychoactive effects, the cannabinoid tetrad was performed on day 15, after the von Frey test. The values obtained for each parameter after a clearly psychoactive dose of WIN (5 mg/kg, ip [20]) were used as reference (see dotted lines in Figure 6). Despite its anti-allodynic effect in the von Frey test, the systemic dose of WIN used in the present study (1 mg/kg, ip) did not induce significant antinociception (Figure 6A), hypothermia (Figure 6C) or hypolocomotion (Figure 6D) in either control (saline) or neuropathic (5-FU) rats, and only induced a significant increase in the time taken for the saline-treated rats to get down the ring in the catalepsy test, although this effect was negligible if compared with the reference values for this parameter (Figure 6B).

### 2.6. Visceral Sensitivity

The evaluation of visceral sensitivity was performed in a separate group of animals 5 days after 5-FU administration, when, as mentioned above, there was significant ileal mucositis and mechanical tactile allodynia, but not colonic mucositis or diarrhea. First of all, after 20 min of cannabinoid administration, the threshold for mechanical stimulation was measured using the von Frey test, as a positive control for the effects previously demonstrated (Figure 5): the pro-neuropathic effect of 5-FU and the anti-allodynic effect of WIN. Indeed, as shown in Figure 7A, animals treated with 5-FU showed a significant decrease in the threshold for mechanical sensitivity versus control animals (*p* < 0.01), confirming the presence of tactile allodynia, and WIN (1 mg/kg, ip) reversed this effect (*p* < 0.001).

Also, the ring test was performed to detect cannabinoid-induced catalepsy (as one of the most relevant signs suggestive of cannabinoid psychoactive effects in the cannabinoid tetrad), to confirm that also in these other animals there was a lack of psychotropic effects, as shown in the previous experiments (Figure 6). The values obtained in the different groups of animals were similar (without statistically significant differences amongst them) and low, far from those obtained in previous studies using a psychoactive dose of WIN (5 mg/kg [25]), confirming our hypothesis (Figure 7B).

Afterwards, visceral sensitivity was measured as abdominal contractions in response to intracolonic tonic mechanical stimulation. Before pressure application, control rats (saline-treated animals that received WIN vehicle, Sal+Veh group) presented some contractions (0.75 ± 0.15 contractions/min). Then, increasing intracolonic pressure evoked an increase in the number of contractions per minute up to 8.85 ± 1.72 contractions/min at the highest pressure used (75 mmHg), indicating sensitivity to colonic mechanical stimulation. When pressure was returned to zero, no contractions were detected (Figure 7C). In this group of animals, abdominal contractions were around 1.2–1.3 s long, regardless of the pressure applied (Figure 7D), except for initial and final intervals, where the pressure applied was 0, and the durations of contractions were around 0.5 and 0 s, respectively. Finally, the percentage of time that animals remained contracting the abdomen progressively increased in a pressure-dependent manner, reaching maximum values of 20.8 ± 4.47% for the highest pressure (75 mmHg) (Figure 7E).

In rats treated with 5-FU and WIN vehicle (5-FU+Veh group), there was also a progressive increase in the number of abdominal contractions (Figure 7C) and percentage of time in contraction (Figure 7E) with mechanical stimulation, and the graphs closely overlapped with those from the control group. Nevertheless, at the highest pressures (60 and 75 mmHg) there was a slight increase in both parameters, without reaching statistically significant differences. Regarding the duration of contractions, the animals treated with 5-FU showed slightly higher values (1.2–1.5 s) compared with the control group. Furthermore, the duration of the contractions at the initial 0 pressure was the highest of all groups (1.2 ± 0.39 s), reaching statistical differences with the control group (*p* < 0.05).

Treatment with the cannabinoid agonist WIN in both saline- (Sal+WIN group) and 5-FU-treated animals (5-FU+WIN group) tended to reduce the number of contractions per minute (Figure 7C) and percentage of time in contraction (Figure 7E) compared with both controls and animals treated only with 5-FU, and, interestingly, this reduction was more intense in the case of animals treated with 5-FU. Moreover, WIN prevented the 5-FU-induced increase in contraction duration at the initial 0 pressure, the difference with control being statistically significant (Figure 7D). Importantly, the differences found for percentage of time in contraction between 5-FU+veh and 5-FU+WIN groups were statistically significant at 45 mmHg (*p* < 0.05), highlighting the antinociceptive effect of this non-psychotropic dose of WIN also in the context of 5-FU hypersensitivity to intracolonic mechanical stimulation.

## 3. Discussion

In this study, a single administration of the antitumoral drug 5-FU at 150 mg/kg induced mucositis and other alterations of the gut wall architecture, thermal hyperalgesia, and slight alterations in gastrointestinal motility (but not diarrhea), which were transitory. However, this dose induced persistent mechanical allodynia and alterations of visceral sensitivity. Importantly, the non-selective cannabinoid agonist WIN alleviated peripheral neuropathy and reduced the responses to intracolonic stimulation in 5-FU-treated animals, and these effects were not associated with psychoactive effects, suggesting that they might occur through the activation of the peripheral endocannabinoid system.

In a previous study, 5-FU at 300 mg/kg induced severe alterations of gastrointestinal motility (including radiographically visible diarrhea) and mucositis [30]. In this case, we have characterized the effects of a lower dose (150 mg/kg) that induced mucositis, as in previous reports [8,38], but not diarrhea.

As previously reported, 5-FU at 150 mg/kg reduced body weight and food intake [38,39,40,41], but did not change water intake [38,42]. The body weight reduction could be explained, at least partially, by the associated reduction in food intake, but other factors could also be involved. Maybe, the presence of mucositis could be responsible for some degree of malabsorption, contributing to weight gain reduction. Indeed, 5-FU at 150 mg/kg induced some alterations in the intestinal structure. In addition to general histological damage in both ileum and colon, a significant shortening of the villi, an increase in chromogranin A-positive cells and in the number of neurons per ganglion, and an increase in muscular thickness were apparent in the ileal samples obtained 4 days after 5-FU. These alterations, also found previously by other investigators, were related to inflammation, an increase in the apoptosis index and a decrease in the mitosis index [8,43]. Similar to these previous reports, we also found that 15 days after 5-FU administration these histological alterations were resolved [8,43]. The decreased villi height and the histological damage found 4 days after 5-FU administration could be due to the direct effect of the antitumoral drug on the tissue. The increase in the number of neurons and the thickness of the muscular layers may be hypertrophic changes to compensate the damage produced by 5-FU, in an attempt to maintain gastrointestinal motor function (see below). In this sense, inflammation, vacuolization and neutrophil infiltration could contribute to the increased thickness of the muscle layers in the small intestine [8,30]. In fact, mucositis was associated with diarrhea in other studies using 5-FU at the same dose and route of administration, but with different sexes and/or strains of rats. Thus, compared with mucositis, diarrhea does not seem to be a robust finding, with some rat strains showing vulnerability (female Dark Agouti [40,41]; male Sprague-Dawley rats [42]) and others displaying relative resistance to its development (male Wistar rats: present results and [8]).

In a previous work [30], the acute effect of 5-FU at 150 mg/kg (0–8 h after administration) could be radiographically observed as a delay in gastric emptying, which affected the filling of the other intestinal regions. However, here, 4 and 14 days after treatment, gastrointestinal transit was like that obtained in saline-treated animals. The same dose of 5-FU delayed gastric emptying and the intestinal transit of liquids measured in rats with scintigraphy [8], although in this case gastrointestinal dysmotility remained 15 days after treatment (but was less intense), when mucosal inflammation had been already resolved. These different results regarding gastrointestinal motor function may be attributed to methodological issues. Indeed, Soares et al. [8] evaluated the distribution of a radiolabeled marker in the isolated stomach and small intestine 30 min after gavage, and performed organ bath experiments using muscle strips from gastric fundus and duodenum. In contrast, we used non-invasive radiographic methods, probably less accurate from a physiological point of view compared with scintigraphy, but also more translatable to the clinical situation. Whatever the case may be, gastric dysmotility may be associated to nausea and emesis occurring during 5-FU treatment [44]. As with other chemotherapeutics (i.e., cisplatin), serotonin might be involved in the delay in gastric emptying. In fact, 5-FU at 50 mg/kg increased the release of serotonin in mice, most likely from enterochromaffin cells, elevating plasma serotonin levels [45,46]. Interestingly, the number of mucosal cells positive for chromogranin A (a marker of the enterochromaffin cells, which release serotonin [47,48]) tended to increase in 5-FU-treated animals compared with controls, and this effect was relatively persistent.

Peripheral neuropathy is a relevant dose-limiting side effect of cancer chemotherapy (including platinum derivatives, vincristine or bortezomib, among others [49]), which highly impairs the patients’ quality of life, even long after treatment cessation [50]. There are very few data in patients treated with 5-FU describing alterations in tactile sensitivity. Only a few articles describe the occurrence of paresthesia in patients treated with this antitumoral drug, but often patients are treated in a multiple schedule with other chemotherapeutics [17,18]. In animals, the repeated utilization of other chemotherapeutic drugs, such as cisplatin [51], paclitaxel [31], oxaliplatin [52] or vincristine [53], induced painful peripheral neuropathy. Furthermore, alterations of mechanical sensitivity induced by the repeated administration of 5-FU have been described to occur in the oral region, associated with oral mucositis [54]. However, to our knowledge, this is the first time that tactile mechanical allodynia and thermal hyperalgesia have been described associated with the use of a single administration of 5-FU at 150 mg/kg. Interestingly, the effect of the antitumoral drug on mechanical sensitivity was persistent, remaining at least up to 15 days after the administration of 5-FU, but the effects on thermal sensitivity were not as robust and, although 3 days after administration a slight thermal hyperalgesia was apparent, it disappeared 15 days after treatment.

An additional important result of our study is that, compared with control animals, those treated with 5-FU showed a relatively increased visceral sensitivity in response to mechanical intracolonic stimulation. This was measured 5 days after 5-FU administration, when mucositis was probably still present (it was evident on day 4 after treatment), but neither gastric dysmotility nor diarrhea were apparent. At that time point, 5-FU-treated animals tended to display an increased duration of abdominal contractions (and percent time contracting the abdomen). The difference with control animals was significant at the initial pressure (0 mmHg), which indicates a higher sensitivity even to the insertion of the balloon in the colon, suggesting that not only tactile but also colonic mechanical allodynia had been developed in these animals at this time point. Animal studies on antitumoral-induced visceral sensitivity are very scarce and methodologically different. For example, in a previous study [21], a high dose of cisplatin (6 mg/kg) significantly decreased the response to intracolonic mechanical stimulation, but in that case, the experiment was performed just 2 h after administration, and the reduced nociceptive response was probably associated with the well-known cisplatin-induced acute release of serotonin and the subsequent vagally-mediated duodenal release of anandamide, which may reduce pain via the activation of the CB1 receptor [55]. In the same way, the acute administration of paclitaxel dose-dependently induced visceral pain in the rat up to 100 min after administration, and this effect was mediated by TRPV1 receptors [22]. Interestingly, as highlighted in these studies, the endocannabinoid system seems to be involved in the development of short-term altered visceral sensation. In our present conditions, the increased response to intracolonic mechanical stimulation seems to be more closely associated with a general, and more permanent, neurotoxic effect of 5-FU underlying both somatic (present results) and, probably also important to visceral nociception, enteric neuropathy [13,56]. It will be interesting to determine if other antitumoral drugs also induce visceral hypersensitivity several days after treatment, resembling peripheral neuropathy, or if this effect is unique to 5-FU.

Finally, the effect of WIN was analyzed on both somatic and visceral nociceptive thresholds, altered by 5-FU. Regarding chemotherapy-induced somatic peripheral neuropathy, cannabinoids were effective in reducing tactile hypersensitivity when applied either systemically or locally [31,32,33,34]. Accordingly, the cannabinoid agonist WIN alleviated the signs of peripheral neuropathy (mechanical allodynia) in the animals treated with 5-FU, and both routes of administration (intraplantar and intraperitoneal) were effective to achieve this, without exerting significant psychoactive effects. Furthermore, the same non-psychotropic dose of WIN used systemically tended to reduce the responses to intracolonic mechanical stimulation in both saline- and 5-FU-treated rats, and these effects were significant for the comparison between the two groups treated with 5-FU, i.e., those that received vehicle vs. those receiving WIN. Thus, WIN was effective in reducing not only tactile allodynia, but also colonic hypersensitivity induced by 5-FU. Although further research is warranted to determine the mechanisms involved, our results support the interest in studying the effects of cannabinoids (either centrally acting at low doses, or peripherally restricted) in neuropathic conditions, because these treatments could reduce the bothersome signs of neuropathy, without exerting the worrying psychotropic effects typical of this type of drug.

Indeed, the effect of the synthetic cannabinoid WIN on somatic and visceral sensitivity suggests that the endocannabinoid system, in particular the peripheral endocannabinoid system, may be altered by chemotherapy, which needs to be more deeply evaluated. Thus, it will be very interesting to determine the involvement of the endocannabinoid system in the visceral and somatic sensitivity alterations caused by 5-FU and possibly other antitumoral drugs, such as cisplatin [21] or paclitaxel [22], at different time points. Targeted drugs (selective agonists of CB1 and/or CB2 receptors, antagonists, etc.) and drugs that modify endocannabinoid levels should be used for this purpose. Finally, it would also be interesting and necessary to study the changes induced by chemotherapy in the molecular expression of the endocannabinoid system components in the enteric nervous system (in the different gastrointestinal organs), and in regions of interest in the central nervous system (brain, spinal cord) and peripheral nervous system (dorsal root ganglia, skin afferents), using both immunohistochemical methods and molecular biology techniques.

In conclusion, the activation of the peripheral endocannabinoid system may be useful to alleviate not only neuropathic, but also visceral pain associated with antitumoral drug treatments.

## 4. Materials and Methods

The study was designed and tests were performed according to the European and Spanish legislation on care and use of experimental animals (2010/63/UE for animal experiments; Real Decreto 53/2013) and received the required institutional approval (PROEX 061/18). All efforts were made to reduce the number of animals used and their suffering. The NIH Guide for the Care and Use of Laboratory Animals was applied at all stages of the research.

### 4.1. Animals and Treatments

Male Wistar rats (250–300 g) were obtained from our Veterinary Unit and housed (4/cage) in transparent cages (60 cm × 40 cm × 20 cm), under regulated conditions (temperature = 20 °C; humidity = 60%), with a 12 h light/12 h dark cycle (lights on at 8 am). Animals were fed ad libitum with standard laboratory rat chow (Harlan Laboratories Inc. Barcelona, Spain) and tap water.

Rats received one intraperitoneal (ip) injection of 5-FU (150 mg/kg) or saline (control group, 6–8 mL/kg). In previous reports, this dose induced mucositis [39] and acute gastrointestinal dysmotility [30], without causing lethality. The different parameters evaluated along the 15 days of the experiment are summarized in Figure 8. A first batch of rats was used to study gastrointestinal motility and the development of neuropathic pain along the 15 days of the study; tissue samples were collected afterwards for histology (day 15). A second batch was used to obtain histological preparations 4 days after 5-FU administration. Finally, a third batch was used to study visceral sensitivity on day 5 after 5-FU treatment.

The first and third batches of animals were used to evaluate the effect of the non-selective cannabinoid agonist WIN 55,212-2 (WIN) on the nociceptive thresholds. Thus, at specified time points (see below), WIN was administered at doses (and routes) known to exert antinociceptive somatic effects without producing significant central effects [31,32,33,34,57]: 50 and 100 µg, intraplantar (ipl); 1 mg/kg, ip. The same systemic dose (1 mg/kg, ip) was assessed in the visceral pain experiments (see below).

**Figure 8 ijms-24-14430-f008:**
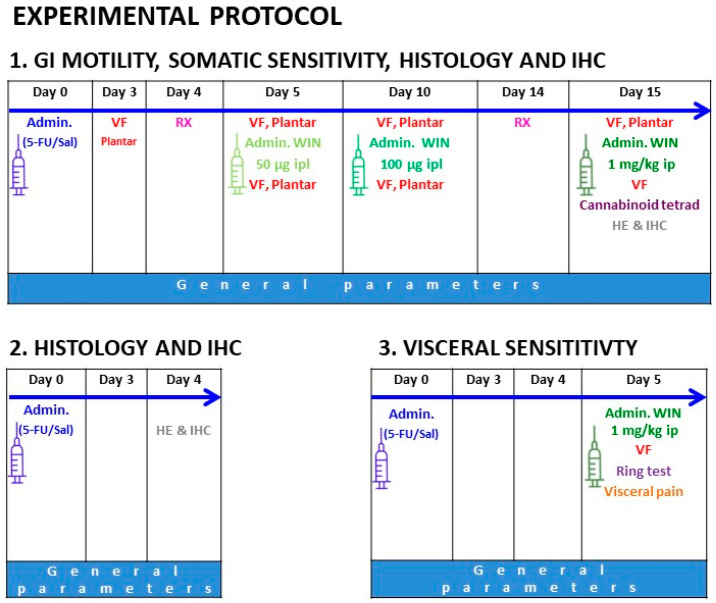
Experimental protocol. The antitumoral drug 5-fluorouracil (5-FU, 150 mg/kg) was administered intraperitoneally (ip) on day 0 to the three different batches of animals to study: (**1**) gastrointestinal (GI) motility, using radiographic methods, as well as mechanical and thermal tactile sensitivity along the 15 days of the experiment, and histology and immunohistochemistry (IHC) on day 15 after 5-FU; (**2**) histology and IHC in gut wall samples, on day 4 after 5-FU; (**3**) visceral sensitivity, on day 5 after 5-FU. Batches 1 and 3 were also used to determine the effects of the cannabinoid agonist WIN 55,212-2 (WIN), at different doses and through different routes, on the nociceptive thresholds; when used systemically (ip), its central effects were measured using the cannabinoid tetrad (batch 1) or the ring test (batch 3). Abbreviations: 5-FU, 5-fluorouracil; Admin, administration; HE, hematoxylin–eosin staining; IHC, immunohistochemistry; ip, intraperitoneal; ipl, intraplantar; Plantar, plantar test; RX, Radiographic study of GI transit; Sal, saline; VF, von Frey hair test; WIN, WIN 55,212-2.

### 4.2. General Health Parameters

Body weight and food and water intakes were evaluated throughout the experiment every 2–3 days. To calculate food and water intake, the amount of chow or water consumed per cage was measured and the average intake per animal and day was calculated.

### 4.3. Gastrointestinal Motility Experiments

Gastrointestinal motility was measured on days 4 and 14 after 5-FU administration using radiographic methods [30]. Briefly, 2.5 mL of barium sulfate (Barigraph^®^ AD, Juste SAQF, Madrid, Spain; 2 g/mL temperature = 22 °C) was administered using an orogastric canula. X-rays were taken 0–8 h after barium administration. For this, rats were placed in prone position using an adjustable hand-made plastic clamp. Habituation prior to the test did not alter gut motility. After each radiograph, the rats were released (immobilization was <2 min long). Radiographs were obtained using a CS2100 (Carestream Dental, Madrid, Spain) digital X-ray apparatus (60 kV, 7 mA; 20 ms; focus distance 50 ± 1 cm) and recorded on Carestream Dental T-MAT G/RA film.

Assessment of the X-rays was carried out by a researcher blind to the treatment received by the animals. Each gastrointestinal region was semiquantitatively evaluated using four different parameters: percentage of the region filled with contrast (0–4); intensity of contrast (0–4); homogeneity of contrast (0–2); and sharpness of the gut region profile (0–2). Thus, for each rat, each region scored a total of 0–12 points in each X-ray.

### 4.4. Histopathological Analysis of Gastrointestinal Regions

Samples were obtained from the terminal ileum and colon on days 4 and 15 (N = 4–7 animals per experimental group), fixed in 10% buffered formalin, embedded in paraffin, and cut into 5 μm-thick sections that were stained with hematoxylin-eosin (HE) or processed for immunohistochemistry (IHC). Sections were studied with a Zeiss Axioskop 2 microscope equipped with the AxioVision 4.6 image analysis software package to calculate morphometric parameters. The analysis was performed by triplicate in up to 8 random fields measured on 20–40× objective microphotographs per section and specimen. The experimenter was unaware of the treatment received by the animals.

HE-stained sections were used for general gut wall structure evaluation. Ileum samples were evaluated according to criteria adapted from Galeazzi et al. (1999) [58]. A numerical score of 0 to 9 was given to each sample, considering the general loss of mucosal structure (0 to 3, from normal to severe), the extent of the inflammatory cellular infiltrate (0 to 3, from absent to transmural), the presence of abscesses in the crypts (0 to 1, from absent to present), the absence of goblet cells (0 to 1, from absent to present) and the thickness of the muscular layer (0 to 1, from normal to reduced). Similarly, colon samples were assessed according to Saccani et al. (2012) [59]. The total score in this case was 0–13, resulting from the sum of mucosal damage (0 to 3, from normal to severe), inflammatory cell infiltration (0 to 4, from absent to severe), separation of the muscular layer and muscularis mucosa (0 to 2, from normal to severe) and absence of goblet cells (0 to 4, from absent to present).

For IHC, sections were washed with phosphate-buffered saline (PBS) with 0.05% Tween 20 (Calbiochem, Darmstadt, Germany), incubated for 10 min in 3% (*vol/vol*) hydrogen peroxide to inhibit endogenous peroxidase activity and blocked with PBS-BSA (phosphate buffered saline—bovine serum albumin) or 1% calf serum for 30 min to avoid nonspecific binding of the primary antibody. Antigen retrieval was performed in a microwave oven with 10 mM citrate buffer (30 min). Sections were then incubated (overnight, 4 °C) with the following antibodies: mouse monoclonal anti-human chromogranin A (1:800; Thermo Scientific, Waltham, MA, USA), to assess the number of enteroendocrine cells per 500 cells in the epithelium; rabbit polyclonal anti-neuron-specific enolase (NSE; 1:50; Sigma, San Luis, MO, USA), to quantify myenteric neurons. Masvision peroxidase kit (Master Diagnostica, Granada, Spain) was used as a secondary antibody. Samples were counterstained with hematoxylin and coverslips were mounted with Eukitt (O. Kindler GmbH & Co., Freiburg, Germany). As negative control, the slides were incubated without the primary antibody.

### 4.5. Assessment of Somatic Nociceptive Thresholds

To evaluate alterations in tactile sensitivity, on days 3, 5, 10 and 15 after 5-FU administration, the von Frey test and the plantar test were performed to detect mechanical allodynia and thermal hyperalgesia, respectively, as previously described [32].

Thresholds to mechanical non-noxious stimulation were measured using calibrated nylon filaments (2–60 g; Bioseb Instruments, Pinellas Park, FL, USA). The animals were habituated to the test chamber for 10 min, in the 2–3 days before the experiment. Every von Frey filament was applied to the plantar surface of each hind paw for 2–3 s; this process was repeated 5 times with each filament. If the animal withdrew the paw in at least three of five trials, the force exerted by the filament was considered the threshold to tactile stimulation. Mechanical allodynia was defined as a significant decrease in the tactile threshold evoked by mechanical stimuli.

The plantar test was performed to detect heat hyperalgesia using a 37370 plantar test (Ugo Basile, Comerio, VA, Italy). A radiant heat beam was applied to each hind paw and the withdrawal latency was recorded. A cut-off time of 25 s was established to avoid tissue damage. The withdrawal time was evaluated in two trials, 2 min apart, and the mean of both values was used for comparison.

### 4.6. Effect of WIN 55,212-2 on Tactile Sensitivity

To characterize the effect of WIN on tactile sensitivity, the cannabinoid was administered to saline- or 5-FU-treated rats, 20 min before the tests.

On days 5 and 10, WIN was administered intraplantarly. Two doses were tested, 50 (day 5) or 100 μg (day 10), in 25 μL of vehicle (30 µL of Tocrisolve in 1 mL of saline), or 25 μL of vehicle alone. Injections were given to the right hind paw and the left one was used as control.

On day 15, the effect of the cannabinoid systemically administered was evaluated. For this, WIN (1 mg/kg) or its vehicle (1 mL/kg) was ip injected and von Frey and plantar tests were carried out 20 min after.

### 4.7. Cannabinoid Tetrad

The cannabinoid tetrad evaluates antinociception (thermal sensitivity), catalepsy, rectal temperature, and spontaneous locomotor activity [60]. To evaluate the central effect of WIN, 5-FU- or saline-treated rats were injected on day 15 with vehicle or WIN (1 mg/kg, ip). The threshold for somatic mechanical stimulation was measured 20 min after cannabinoid administration, and after that, the cannabinoid tetrad was measured as previously reported [38].

Heat-antinociception was tested 25 min after drug administration, as described above.

The ring test was used to measure catalepsy; animals were suspended by their front paws from a 20 cm-high glass; this height allowed the rat to just touch the surface with their hind paws. A cut-off limit of 30 s was imposed, and the time taken for the rat to move off the ring was measured. Latencies were measured 30 min after drug or vehicle administration.

Core temperature was measured with a P6 thermometer and a rectal probe (Cibertec S.A., Madrid, Spain) introduced into the rectum at 5 cm. Temperatures were recorded 35 min after WIN or vehicle administration.

Spontaneous locomotor activity was measured 40 min after drug administration using an actimeter (Cibertec S.A.). To do this, animals were placed in the recording chambers provided with photocell beams; the number of interruptions of these photocells was recorded for 30 min.

### 4.8. Assessment of Visceral Sensitivity

Colorectal sensitivity was measured on day 5 after saline or 5-FU, in a separate group of animals, as previously described [21]. In these animals, general parameters (body weight and food and water consumption) were also recorded on day 3 and on day 5, before the experiment.

Rats were sedated with Sedator^®^ (medetomidine hydrochloride, 1 mg/kg, ip). To facilitate the visualization of abdominal contractions during the experiment, a 10 cm longitudinal line was drawn over the linea alba of the abdomen with transverse lines every 2 cm. Feces was removed from the rectum and a latex balloon (5 cm long) lubricated with vaseline was inserted to a depth of 7 cm inside the colorectum. The balloon was connected to a catheter and fixed to the tail of the rat with Parafilm^®^. The catheter was connected to a sphygmomanometer.

Revertor^®^ (atipamezole hydrochloride, 0.66 mg/kg, ip) was used to revert sedation. An iPad (Apple, Madrid, Spain) was placed 30 cm below the recording chamber and once the rats were awake, their behavior was recorded for 40 min. The first 5 min of the recording were only used to confirm the recovery from sedation and were discarded. Thereafter, the sphygmomanometer was used to gradually increase the pressure of the intracolonic balloon, from 0 to 75 mmHg, in steps of 15 mmHg every 5 min, and finally returned to 0 mmHg again (a single stimulus was applied for each pressure value and maintained for 5 min (tonic stimulation protocol [21])).

The videos were exported at 1 frame/s, using Quick Time Player Pro for Windows (v.7.7.4; Apple Inc.). Abdominal contractions in response to mechanical intracolonic stimulation were analyzed to evaluate visceral sensitivity. An abdominal contraction was considered as a depression of the abdomen where transverse lines approached one another. The average number (per min) and duration of contractions were determined, as well as the average percentage of time spent by the rat contracting the abdomen during each 5 min period.

### 4.9. Effect of WIN 55,212-2 on Visceral Sensitivity

To characterize the effect of WIN on visceral sensitivity, the cannabinoid or its vehicle were administered to saline- or 5-FU-treated rats, at 1 mg/kg (ip). Mechanical sensitivity (von Frey test) and catalepsy (ring test) were measured 20 and 25 min, respectively, after administration, to confirm the presence of somatic neuropathy induced by 5-FU, and the lack of psychotropic effects of this dose of WIN in this cohort of both control (saline) and neuropathic (5-FU) animals. Thereafter, colorectal sensitivity was evaluated as described above.

### 4.10. Compounds and Drugs

Barium sulfate (Barigraf^®^ AD, Juste SAQF) was suspended in tap water and continuously hand-stirred until administration. 5-FU and WIN 55,212-2 were purchased from Sigma-Aldrich (Madrid, Spain), and suspended in saline (sonicated for about 1.5 h) and Tocrisolve (Tocris, Cookson, Bristol, UK), respectively.

### 4.11. Statistical Analysis

Data were expressed as mean values ± the standard error of the mean (SEM). Differences between groups were analyzed using one- or two-way ANOVA followed by post-hoc Bonferroni’s multiple comparison test. Values of *p* < 0.05 were regarded as significantly different. Statistical analyses were done using Prism 8.0 (GraphPad Software Inc., La Jolla, CA, USA).

## Figures and Tables

**Figure 1 ijms-24-14430-f001:**
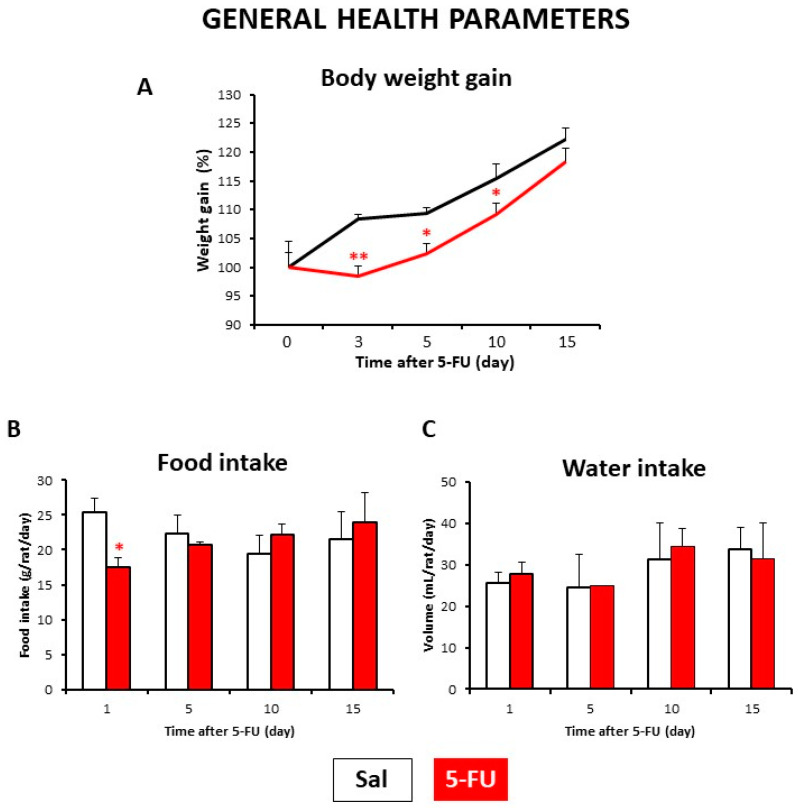
Effect of the antitumoral drug 5-fluoruracil on general health parameters in the rat. Body weight gain (**A**) and food (**B**) and water intake (**C**) were evaluated in control animals treated with saline (Sal, black line or white bars) or 5-fluoruacil (5-FU, 150 mg/kg, ip; red line or bars). Data represent mean ± SEM. N = 8 each group (two-way ANOVA followed by Bonferroni’s post-hoc test). * *p* < 0.05, ** *p* < 0.01 vs. saline.

**Figure 2 ijms-24-14430-f002:**
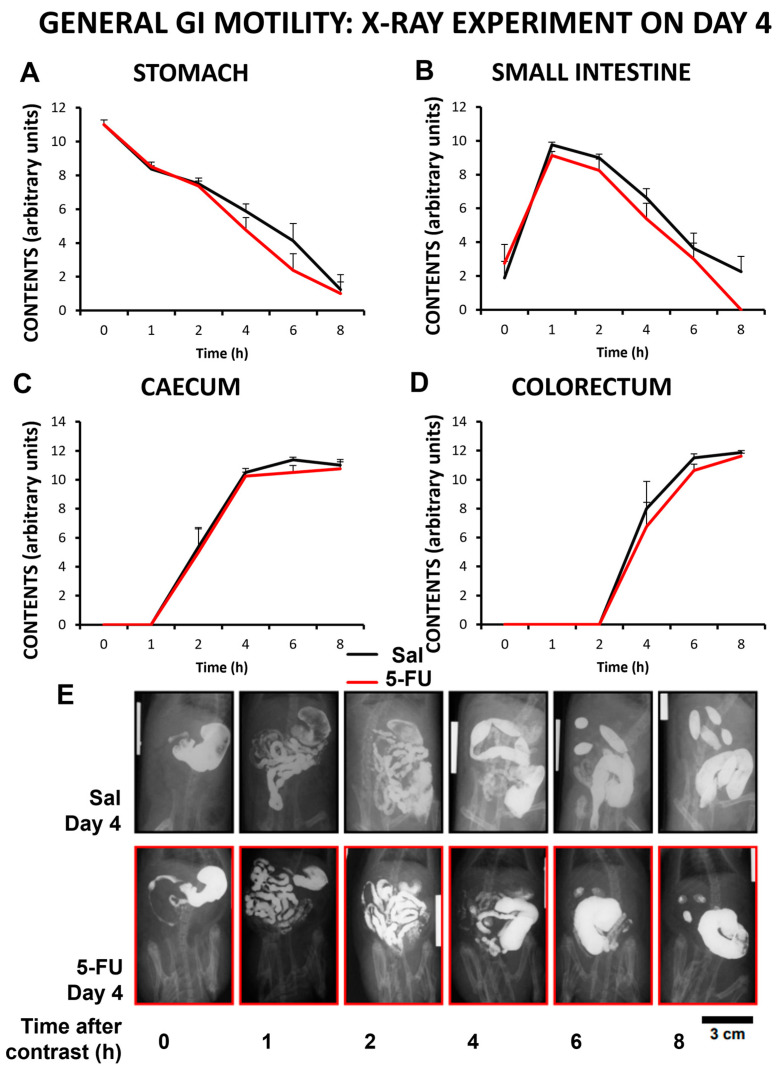
Effect of the antitumoral drug 5-fluororuacil on gastrointestinal motor function in the rat. Motor function was evaluated using radiological methods (see text) in rats treated with saline (Sal, black line) or 5-fluorouracil (5-FU, 150 mg/kg, ip; red line) on day 4 after 5-FU. At that time point, barium sulfate (2.5 mL, 2 g/mL) was gavaged, and radiographs were taken 0–8 h after contrast administration. Data represent mean ± SEM for motor function in stomach (**A**), small intestine (**B**), cecum (**C**), and colorectum (**D**). N = 8 each group (two-way ANOVA followed by Bonferroni post-hoc test). (**E**) Representative radiographic images obtained for the different treatment groups. Scale bar: 3 cm.

**Figure 3 ijms-24-14430-f003:**
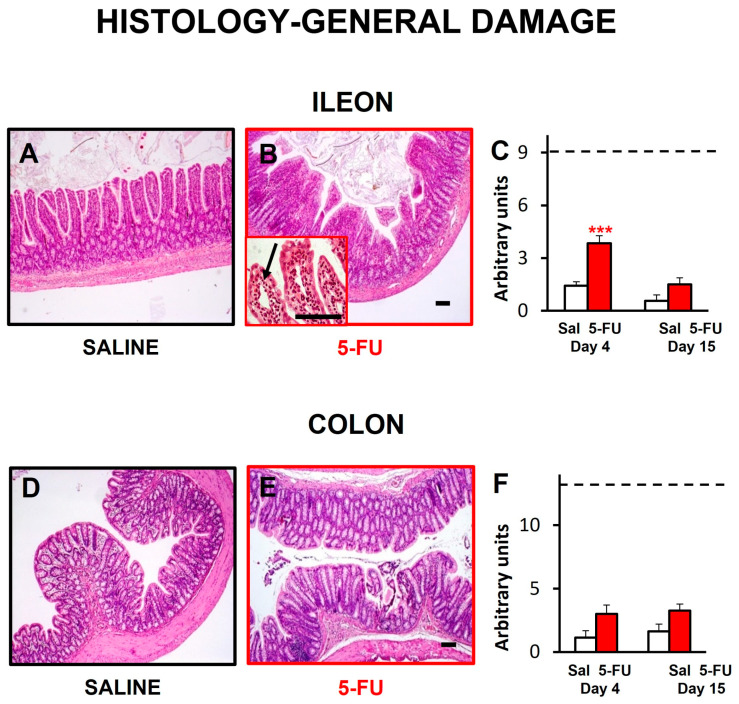
General study of the histological damage of small and large intestine after 5-FU administration in the rat. Ileum (**A**,**B**) and colon (**D**,**E**) micrographs showing organ architecture in animals treated with saline (Sal; (**A**,**D**)) or 5-fluorouracil (5-FU, 150 mg/kg, ip; (**B**,**E**)); these images correspond to samples obtained on day 4 after treatment. Insert in (**B**) shows enlarged lymph vessels (arrow). Bar: 100 µm. Quantitative analyses of the histological damage are shown for ileum (**C**) and colon (**F**). Bars represent mean values ± SEM for Sal-treated (control, white) and FU-treated animals (red), 4 or 15 days after 5-FU administration. Dotted lines indicate the maximum damage score that could be reached in the ileum (**C**) and colon (**F**). N = 4–7 each group. *** *p* < 0.001 vs. saline (two-way ANOVA followed by Bonferroni’s post-hoc test).

**Figure 4 ijms-24-14430-f004:**
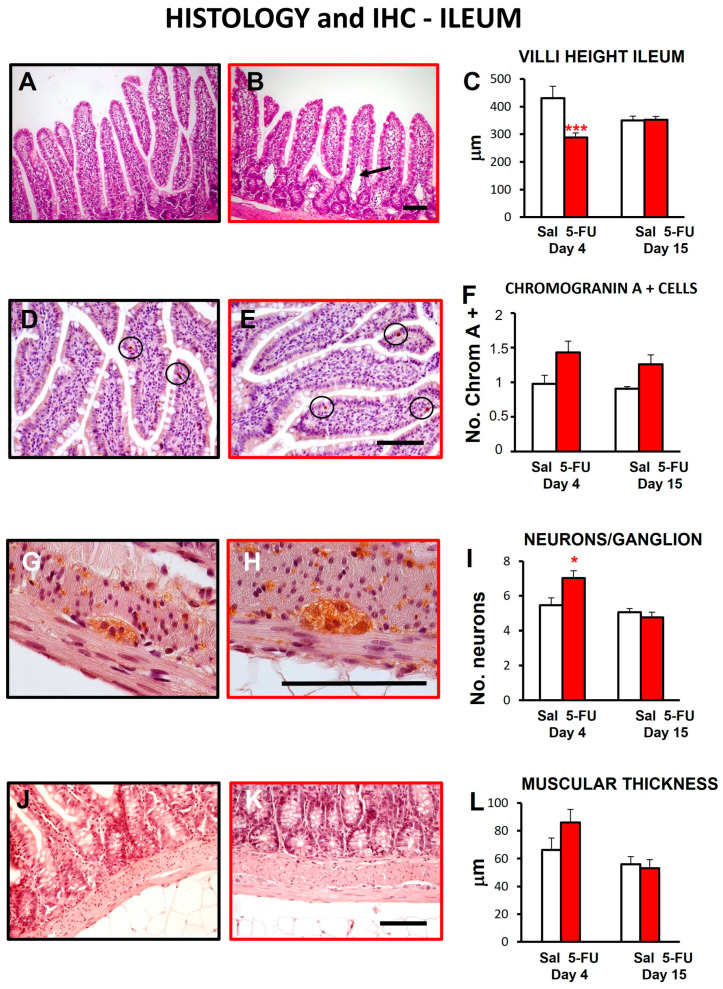
Specific histological and immunohistochemical study of the effect of 5-FU on the rat ileum**.** Micrographs showing the structural ileum features in saline-treated animals (Sal; (**A**,**D**,**G**,**J**)) or 5-fluorouracil (5-FU, 150 mg/kg, ip; (**B**,**E**,**H**,**K**)), 4 days after treatment. Ileal villi (**A**,**B**) with enlarged lymph vessels (arrow in (**B**)), chromogranin A+ cells in mucosa (encircled in (**D**,**E**)), neurons in the myenteric ganglia (**G**,**H**), and muscle layers (**J**,**K**). Bars: 100 µm. Quantitative analyses of the histological samples (**C**,**F**,**I**,**L**). Bars represent mean values ± SEM for Sal-treated (control, white) and 5-FU-treated animals (red), 4 or 15 days after 5-FU administration. N = 4–7 each group. * *p* < 0.05, *** *p* < 0.001 vs. saline (two-way ANOVA followed by Bonferroni’s post-hoc test).

**Figure 5 ijms-24-14430-f005:**
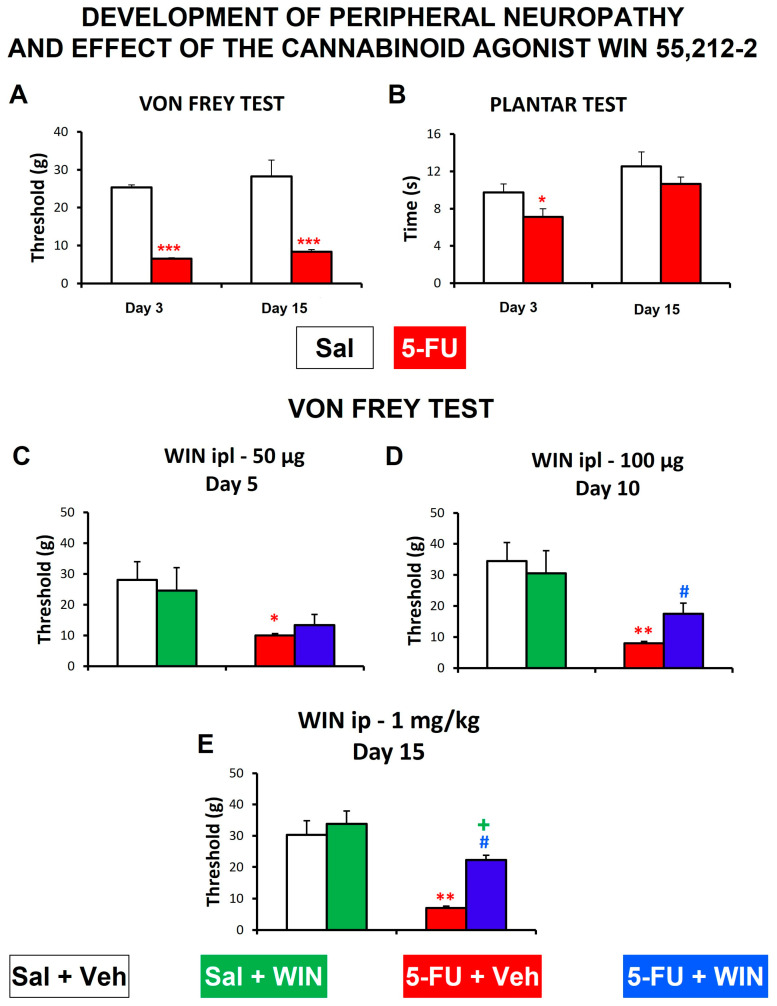
Development of peripheral neuropathy induced by the antitumoral drug 5-fluorouracil and effect of the cannabinoid agonist WIN55,212-2 locally or systemically administered in the rat. Saline (Sal, white bars) or 5-fluorouracil (5-FU, 150 mg/kg, ip; red bars) were administered and the thresholds for mechanical sensitivity (**A**) and thermal sensitivity (**B**) were evaluated with the von Frey test and the plantar test, respectively. The cannabinoid agonist WIN 55,212-2 (WIN) or its vehicle (Veh) was locally (intraplantarly, ipl: 50 µg (**C**); 100 µg (**D**)) or systemically (intraperitoneally, ip: WIN 1 mg/kg, (**E**)) administered at different time-points after 5-FU (5, 10 and 15 days, respectively), and the effect was thereafter evaluated using the von Frey hair test. Bars represent mean ± SEM. N = 8 each group. * *p* < 0.05, ** *p* < 0.01, *** *p* < 0.001 vs. control (Sal or Sal+Veh); # *p* < 0.05 vs. 5-FU+Veh; + *p* < 0.05 vs. Sal+WIN (one or two-way ANOVA followed by Bonferroni’s post-hoc test).

**Figure 6 ijms-24-14430-f006:**
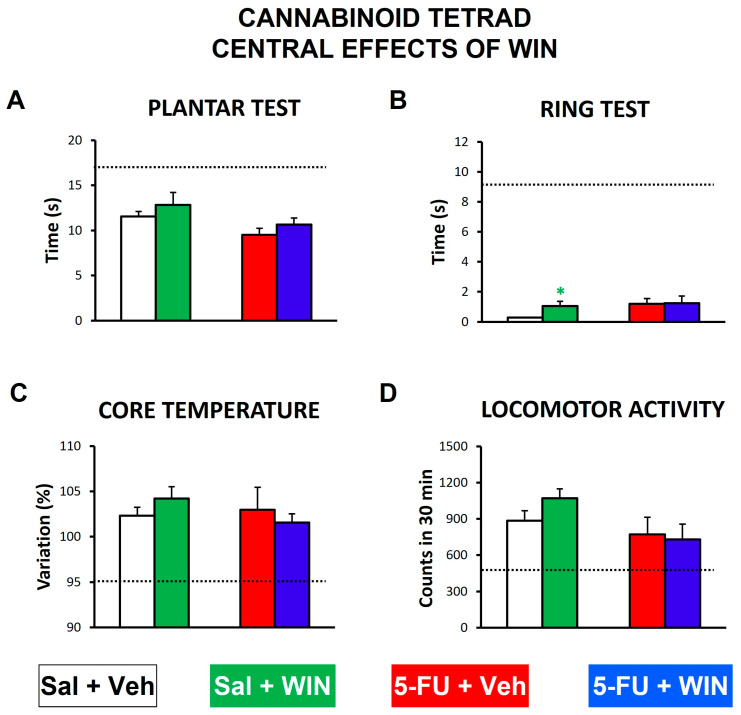
Central effects of the cannabinoid agonist WIN 55,212-2 systemically administered in the rat. Cannabinoid central effects were evaluated using the cannabinoid tetrad: (**A**) plantar test (for antinociception); (**B**) ring test (for catalepsy); (**C**) core temperature (for hypothermia); (**D**) locomotor activity (for hypolocomotion). Tests were performed as described in the text. At 15 days after saline (Sal) or 5-fluorouracil (5-FU, 150 mg/kg, ip) administration, WIN 55,212-2 (WIN, 1 mg/kg, ip) or its vehicle (Veh) were administered, and the cannabinoid tetrad was performed 20 min after. Bars represent mean ± SEM. N = 8 animals each group. * *p* < 0.05, vs. Sal+Veh (one-way ANOVA followed by Bonferroni’s post-hoc test). Dotted lines indicate the values of reference obtained in each test by a higher, psychoactive dose of the cannabinoid (WIN 5 mg/kg [32]).

**Figure 7 ijms-24-14430-f007:**
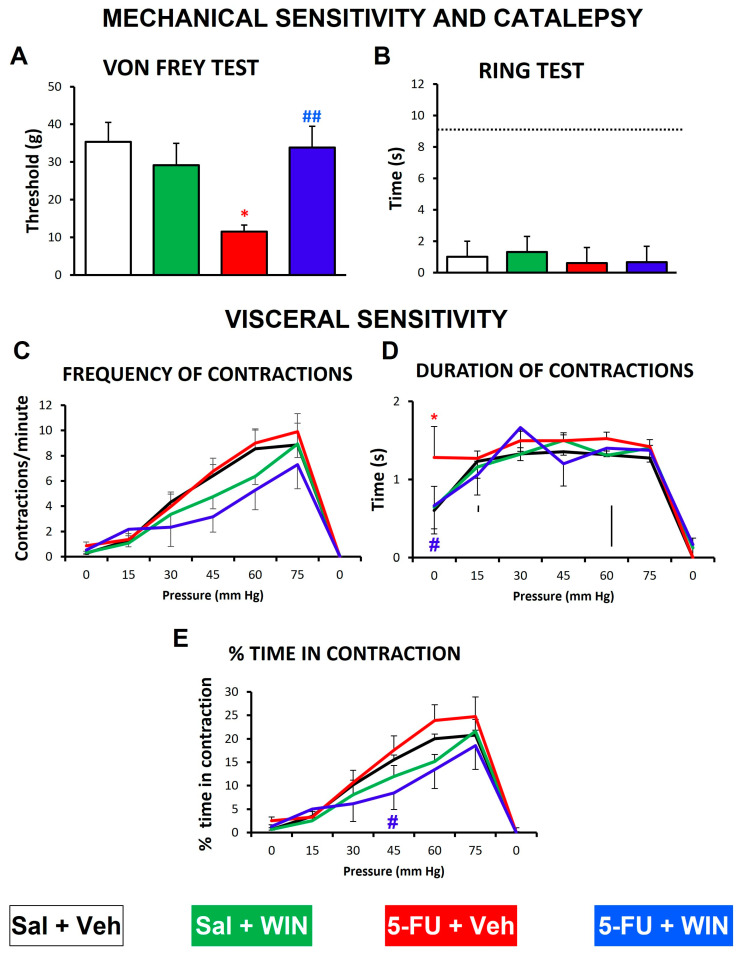
Effect of the antitumoral drug 5-fluororuacil and/or the cannabinoid agonist WIN 55,212-2 on responses to intracolonic mechanical stimulation in the rat. After 5 days of saline (Sal) or 5-fluorouracil (5-FU, 150 mg/kg, ip) administration, the rats were injected with WIN 55,212-2 (WIN, 1 mg/kg, ip) or its vehicle (Veh). Mechanical sensitivity and catalepsy were evaluated using the von Frey test (**A**) and the ring test (**B**), respectively, 20 min after injections. Then, the animals were prepared for the evaluation of visceral sensitivity by inserting in the colon a balloon connected to a sphygmomanometer, which allowed for mechanical intracolonic stimulation (see text). A tonic stimulation protocol was used: pressure was increased from 0 to 75 mmHg, in steps of 15 mmHg every 5 min, to finally return to 0 mmHg again; each pressure stimulus was applied only once and maintained for 5 min. The number of abdominal contractions per minute (**C**), the average duration of the abdominal contractions (**D**) and the % of time in contraction (**E**) were measured. Data represent the mean ± SEM. N = 8 each group. * *p* < 0.05 vs. Sal+Veh; # *p* < 0.05; ## *p* < 0.01 vs. 5-FU+Veh (one or two-way ANOVA followed by Bonferroni’s post-hoc test). The dotted line in (**B**) represents the value of reference obtained in the ring test by a higher, psychoactive dose of the cannabinoid (WIN 5 mg/kg [32]).

## Data Availability

Data will be available upon reasonable request to the corresponding author (R.A.).

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
