# Peer review of "Effect of the Cannabinoid Agonist WIN 55,212-2 on Neuropathic and Visceral Pain Induced by a Non-Diarrheagenic Dose of the Antitumoral Drug 5-Fluorouracil in the Rat"

_ijms, 2023, doi:10.3390/ijms241914430_

Round 1

Reviewer 1 Report

In this study the authors show interesting results. Importantly, the cannabinoid agonist significantly decreases the allodynia induced by 5-FU. 

However, in the results where they prove that the cannabinoid has no central effects, the authors use the plantar test 15 days after having administered 5-FU, when in Figure 5B they proved that with this test at 15 days no thermal hyperalgesia was observed.

I suggest using the test in which antinociception was observed. 

Author Response

Dear reviewer,

Thank you very much for your comments on our manuscript. Please see attached our responses.

Kind regards,

Dr Raquel Abalo

Reviewer 2 Report

This paper has been submitted to the planned special issue of IJMS entitled "The Endocannabinoid System: New Insights into Its Role in Health and Disease." The aim of the authors was to evaluate the effects of the cannabinoid agonist WIN 55,212-2 on neuropathic and visceral pain induced by a non-diarrheagenic dose of the antineoplastic drug 5-fluorouracil.  What was the reason that only the WIN agonist was investigated? WIN is a synthetic cannabinoid compound, but it is not part of the endocannabinoid system.  Which cannabinoid receptor is involved in mediating the pain-relieving effects of WIN could and should have been studied using cannabinoid and even opioid receptor antagonist ligands.  The experimental background of the work is indeed colorful and complex, but the reviewer is not convinced, e.g. about the correctness of the selection of the treatment doses used or the timing of the applied protocols.  I don't really understand the presentation of the histological and immunohistochemical analysis, since the cannabinoid title compound WIN was not involved at all in this experimental setup.  In my opinion, the publication of the article requires a significant revision.

„Közösség által ellenÅ‘rizve” ikon

Author Response

(The authors gave the same response as above.)

Reviewer 3 Report

Dear Authors, 

The draft you presented reports the results of the co-administration in vivo of the antitumor widely used 5FU and a cannabinoid agonist. The authors' aim was to investigate several effects on rat models of 5FU and how they would have been altered by the co-administration with the cannabinoid agonist. 

I can say the study is well designed, the aim and the methods used are clearly described, and the limits of the techniques used were also reported next to the pros. The authors depicted and graphed the results and commented on the conducted assays and the obtained results in a rational and linear way. 

The discussion paragraph summed all the aims, what they wanted to evaluate and the obtained data in a clear and straight manner, making it very readable and understandable. The writing and the language are also very good. 

My only suggestion concerns the graphs in most of the pictures. I would suggest improving the labels of some y axes. While in Figure 1 the reader can directly catch what is reported in y axes, figures 2 and 3 are not as clearly readable as previous. Also, Figures 5, 6 and 7 may be improved to be more straightforward and understandable. Figure 5, panel E is not labelled, it should be corrected. 

Best wishes

Author Response

(The authors gave the same response as above.)

Round 2

Reviewer 2 Report

Since the authors gave satisfactory answers to the questions and comments I raised, I recommend the acceptance and publication of the article.